# Orchestrating privacy-protected big data analyses of data from different resources with R and DataSHIELD

**Yannick Marcon**[1]*, **Tom Bishop**[2], **Demetris Avraam**[3], **Xavier Escriba-Montagut**[4,5], **Patricia Ryser-Welch**[3], **Stuart Wheater**[6], **Paul Burton**[3], **Juan R. González**[4,5,7,8]*

**1** Epigeny, St Ouen, France, **2** MRC Epidemiology Unit, University of Cambridge, Cambridge, United Kingdom, **3** Population Health Sciences Institute, Newcastle University, Newcastle, United Kingdom, **4** Barcelona Institute for Global Health (ISGlobal), Barcelona, Spain, **5** Universitat Pompeu Fabra (UPF), Barcelona, Spain, **6** Arjuna Technologies, Newcastle, United Kingdom, **7** Centro de Investigación Biomédica en Red en Epidemiología y Salud Pública (CIBERESP), Barcelona, Spain, **8** Dept. of Mathematics, Universitat Autònoma de Barcelona (UAB), Bellaterra (Barcelona), Spain

* yannick.marcon@obiba.org (YM); juanr.gonzalez@isglobal.org (JRG)

**Data Availability Statement:** Data are available as a figshare project in this link: https://figshare.com/projects/Orchestrating_privacy-protected_big_

## Abstract

Combined analysis of multiple, large datasets is a common objective in the health- and bio-sciences. Existing methods tend to require researchers to physically bring data together in one place or follow an analysis plan and share results. Developed over the last 10 years, the DataSHIELD platform is a collection of R packages that reduce the challenges of these methods. These include ethico-legal constraints which limit researchers' ability to physically bring data together and the analytical inflexibility associated with conventional approaches to sharing results. The key feature of DataSHIELD is that data from research studies stay on a server at each of the institutions that are responsible for the data. Each institution has control over who can access their data. The platform allows an analyst to pass commands to each server and the analyst receives results that do not disclose the individual-level data of any study participants. DataSHIELD uses Opal which is a data integration system used by epidemiological studies and developed by the OBiBa open source project in the domain of bioinformatics. However, until now the analysis of big data with DataSHIELD has been limited by the storage formats available in Opal and the analysis capabilities available in the DataSHIELD R packages. We present a new architecture ("*resources*") for DataSHIELD and Opal to allow large, complex datasets to be used at their original location, in their original format and with external computing facilities. We provide some real big data analysis examples in genomics and geospatial projects. For genomic data analyses, we also illustrate how to extend the resources concept to address specific big data infrastructures such as GA4GH or EGA, and make use of shell commands. Our new infrastructure will help researchers to perform data analyses in a privacy-protected way from existing data sharing initiatives or projects. To help researchers use this framework, we describe selected packages and present an online book (https://isglobal-brge.github.io/resource_bookdown).

data_analyses_of_data_from_different_resources_with_R_and_DataSHIELD/96377. The R code is available at https://isglobal-brge.github.io/resource_bookdown/

**Funding:** This research has received funding from the European Union's Horizon 2020 research and innovation programme under grant agreement No 874583 (ATHLETE) an No 824989 (EUCAN-Connect); the Ministerio de Ciencia, Innovación y Universidades (MICIU), Agencia Estatal de Investigación (AEI) and Fondo Europeo de Desarrollo Regional, UE (RTI2018-100789-B-I00) also through the "Centro de Excelencia Severo Ochoa 2019-2023" Program (CEX2018-000806-S); and the Catalan Government through the CERCA Program. This article is part of the project VEIS: 001-P-001647 co-financed by the European Regional Development Fund of the European Union in the framework of the Operational Program FEDER of Catalonia 2014-2020 with the support of the Secretaria d'Universitats i Recerca del Departament d'Empresa i Coneixement de la Generalitat de Catalunya. This work also forms part of Newcastle University's methods program in Health Data Science addressing the securing of sensitive data; with support from the Department of Health and Social Care under the Connected Health Cities (North East North Cumbria) project and from joint Wellcome Trust/Medical Research Council grant 108439/A/15/Z. The funders had no role in study design, data collection and analysis, decision to publish, or preparation of the manuscript.

**Competing interests:** The authors have declared that no competing interests exist.

## Author summary

Data sharing enhances understanding of research results beyond what is possible from any single study. Data pooling across multiple studies increases statistical power and allows exploration of between-study heterogeneity. But, considerations related to ethico-legal and intellectual/commercial value regularly prevent or impede physical data sharing. DataSHIELD is designed to circumvent this problem. However, despite the growing confidence users have been placing in DataSHIELD to perform privacy-protected analyses of data in cohort consortia, there are real challenges to federated analytics. They include considering the wide range of data formats, and big data sources used, for example, in 'omics-based research. This article describes the development and implementation of the new "resources" architecture in DataSHIELD that overcomes this limitation. We illustrate its value with real world examples related to genomics and geographical data. We also demonstrate how genomic data sharing initiatives such as GA4GH and EGA can benefit directly from our development. Our new infrastructure will help researchers to perform data analyses in a privacy-protected way from existing data sharing initiatives or projects.

This is a *PLOS Computational Biology* Software paper.

## Introduction

Big Data brings new opportunities to biomedicine and challenges to data scientists. These challenges require new computational and statistical paradigms to deal with important principles of data management and data sharing. The new paradigm should consider: ensuring appropriate levels of security and privacy [1]; the rigorous application of the stringent regulations required by governance frameworks such as GDPR in Europe (https://gdpr-info.eu/) and similar regulatory mechanisms across North America and elsewhere; and a considered choice between central data warehousing and the distributed (federated) analysis of data that remain with their custodian[2,3].

Historically there has tended to be a focus on warehousing because of the technical challenges of federation [4], and funder requirements to physically share public data [5]. This requires data custodians to physically transfer data to a central location to make them accessible to analytic users. However, the potential benefits of remote and federated approaches to analysing data are now widely recognised [2,3]. Most fundamentally, the physical data then remain under the control of their custodian with limited access. This can offer major benefits in terms of: making it easier to meet ethics and governance requirements; enhanced flexibility to refine and rerun analyses quickly without waiting for an analyst at each institution to follow an updated analysis plan; allowing datasets to be updated quickly without needing to be resent to a central location [2,6].

Anticipating the future growth of federated analysis, the DataSHIELD (www.datashield.ac.uk) and OBiBa (www.obiba.org) projects are now 10 years into the joint development of an open-source analytics platform that enables and simplifies flexible but efficient federated analysis [2,7,8]. DataSHIELD is linked to an Opal database designed for data management, harmonization and dissemination [9]. In addition, it actively constrains the risk of information disclosure: i.e. the risk that a data analyst is able–accidently or deliberately—to infer individual level data [2,10]. The DataSHIELD platform has a growing user community and a central role

in the analytic strategies of several large research consortia focussed primarily on the federated analysis of large cohort studies, particularly in Europe and Canada. These include BioSHaRE [8], EUCAN-connect [11], LifeCycle [12], ATHLETE [13] and InterConnect [14]. This article describes a radical extension to the DataSHIELD/Opal platform–the *"Resources"* architecture. This allows DataSHIELD to be used in a range of new settings which include the analysis of high-volume data such as omics, geospatial or neuroimaging among others.

Despite the growing confidence users have been placing in DataSHIELD to perform privacy-protected analyses, there have, to date, been several serious limitations: (1) Difficulty in applying DataSHIELD federated analytics across the wide range of data formats, and data sources used in 'omics-based research. (2) Challenges to the efficient porting of high-volume distributed data into the analytic (R) environments on the remote data servers; single large tranches can overwhelm the handling capacity of the system and regular block-by-block refreshments of the analytic data can be impractically slow. (3) To date DataSHIELD has primarily been applied to research settings (typically large cohort studies) where the emphasis has been on the provision of robust disclosure control for analysing sensitive data. But in big data analyses the emphasis is more typically on fast, efficient analysis and data governance often requires a relatively basic level of disclosure control. For example, many consortium-based 'omics projects would like data to remain with their usual generators/custodians–*i.e.* a federated approach avoiding the physical transfer of data to users—and for analysts to be unable to see, copy, capture or otherwise infer, those individual-level data. Crucially, if limitations "1" and "2" could be circumvented it would then be straightforward to ensure that this basic level of disclosure control is embedded into all new functions as they are developed and implemented. This would greatly accelerate the development of new functionality making it realistic to consider rapid implementation of Bioconductor [15], Neuroconductor [16] or R packages designed for big data analyses into DataSHIELD. In one stroke, these objectives have all been realised with the development and implementation of the new **"Resources"** architecture in DataSHIELD/Opal.

This article describes this new facility, illustrates its value with real world examples and considers the exciting implications it has for the future development of the DataSHIELD platform and for the wider adoption of federated approaches to big data analysis. To help researchers use this framework, we present an online book (https://isglobal-brge.github.io/resource_bookdown/) covering installation, sources of help, specialized topics pertaining to specific aspects of privacy-protecting analysis and complete workflows analysing various examples from biomedical, omics and geospatial settings. The packages developed are available through CRAN or Github repositories under open source licenses (GPL3 or MIT).

## Design and implementation

### The resources architecture

When analysing data, it is normal to deal with a very wide variety of data formats, data storage systems and programmatic interfaces. The purpose of the work we describe is not to define a new data format or storage system. Instead we have aimed to describe how data can be accessed, in a formal but generic way, to simplify the integration of various data or computation resources in a statistical analysis program. By actively embracing the variety of data formats and computation systems we seek to guarantee that the right tool can always be used for the type and the volume of data that are being considered.

We define a "resource" to be a description of how to access either: (1) data stored and formatted in a particular way or (2) a computation service. Therefore, the descriptors for the resource will contain the following elements: (1) the location of the data or of the computation

services, (2) the data format (if this information cannot be inferred from the location property) or the format of the function call to the computation service, (3) the access credentials (if some apply).

Once a resource has been formally defined, it becomes possible to build a programmatic connection object that will make use of the data or computation services described. This resource description is not bound to a specific programmatic language (the URL property is a web standard, other properties are simple strings) and does not enforce the use of a specific software application for building, storing and interpreting a resource object. Section 7.8 in our online book describe some examples of resources available in a demonstration Opal repository.

The data format refers to the intrinsic structure of the data. A very common family of data formats is the tabular format which is made of rows (entities, records, observations etc.) and columns (variables, fields, vectors etc.). Examples of tabular formats are the delimiter-separated values formats (CSV, TSV etc.), the spreadsheet data formats (Microsoft Excel, LibreOffice Calc, Google Sheets etc.), some proprietary statistical software data formats (SPSS, SAS, Stata etc.), the database tables that can be stored in structured database management systems that are row-oriented (MySQL, MariaDB, PostgreSQL, Oracle, SQLite etc.) or column-oriented (Apache Cassandra, Apache Parquet, MariaDB ColumnStore, BigTable etc.), or in semi-structured database management systems such as the documented-oriented databases (MongoDB, Redis, CouchDB, Elasticsearch etc.). When the data model is highly structured or particularly complex (data types and objects relationships), a domain-specific data format is sometimes designed to handle the complexity. This then enables statistical analysis and data retrieval to be executed as efficiently as possible. Examples of domain-specific data formats are regularly encountered in the genomic or geospatial fields of research. A data format can also include additional features such as data compression, encoding or encryption. Each data format requires an appropriate reader software library or application to extract the information or perform data aggregation or filtering operations.

Data storage can simply be realised via a file that can be accessed directly from the host's file system or downloaded from a remote location. More advanced data storage systems can involve software applications that expose an interface to query, extract or analyse the data. These applications can make use of a standard programming interface (e.g. SQL) or expose specific web services (e.g. based on the HTTP communication protocol) or provide a software library (in different programming languages) to access the data. These different ways of accessing the data are not mutually exclusive. In some cases when the (individual-level) micro-data cannot be extracted, computation services returning aggregated/summary statistics may be provided. The data storage system can also apply security rules, requiring authentication and proper authorisations to access or analyse the data.

The resource location description will make use of the web standard "Uniform Resource Identifier (URI): Generic Syntax".[17] More specifically, the Uniform Resource Locator (URL) specification is what we need for defining the location of the data or computation resource: the term Uniform allows the resource to be described in the same way, independently of its type, location and usage context; the use of the term Resource does not limit the scope of what might be a "resource", e.g. a document, a service, a collection of resources, or even abstract concepts (operations, relationships, etc.); the term Locator both identifies the resource and provides a means of locating it by describing its access mechanism (e.g. the network location). The URL syntax is composed of several parts: (1) a scheme, that describes how to access the resource, e.g. the communication protocols "https" (secured HTTP communication), "ssh" (secured shell, for issuing commands on a remote server), or "s3" (for accessing Amazon Web Service S3 file store services), (2) an authority (optional), e.g. a server name address, (3) a path

that identifies/locates the resource in a hierarchical way and that can be altered by query parameters.

The resource's data format might be inferred from the path component of the URL; for example, by using the file name suffix. However, it is not always possible to identify the data format because the path could make sense only for the data storage system, for example when a file store designates a document using an obfuscated string identifier or when a text-based data format is compressed as a zip archive. The format property can provide this information. Although the authority part of the URL can contain user information (such as the username and password), it is discouraged to use this capability for security considerations. The resource's credentials property will be used instead, and will be composed of an identifier sub-property and a secret sub-property, which can be used for authenticating with a username/ password, an access token, a key pair (private and public keys), or any other credentials encoded string. The advantage of separating the credentials property from the resource location property is that a user with limited permissions could have access to the resource's location information while the credentials are kept secret.

## The *resourcer* R package

The *resourcer* package is an R implementation of the data and computation resources description and connection. It reuses many existing R packages for reading various data formats and connecting to external data storage or computation servers. The resourcer package's role is to interpret a resource description object to build the appropriate resource connection object. Because the scope of resources is very wide, the *resourcer* package provides a framework for dynamically extending the interpretation capabilities to new types of resources. Next, we describe the key issues to deal with resources within the R environment. Further details and examples are available in Section 7.8 in our online book.

**Resources R Implementation.** The resource class is a simple R structure that holds the properties of the resource described in the previous section: URL, format, identity and secret. To simplify the designation of a resource, an additional name attribute is defined. This identifier is optional and is not necessarily unique. The ResourceClient is a key R6 class, which wraps a resource object and defines operations that can be performed on it. The resourcer package has built-in support for the following use cases:

- Data file resource whose location is defined by the resource's URL and that can be downloaded in a temporary folder to be read. The file locations that are supported by default are: (1) the local file system (obviously with a no-op download), (2) an HTTP(S) connection, optionally providing basic authentication based on the resource's credentials, (3) the MongoDB GridFS file store, (4) Opal's file store and (5) a remote SSH server. For the reading part, the resourcer package uses some tidyverse R packages (such as haven for the SAS, SPSS and Stata data formats, readr for delimited data formats and readxl for Excel data formats) or can load an R object based on its class name specified in the resource's data format property.

- SQL database resource which has a connector based on R's interface for databases (DBI). The resource's URL indicates which database connector is to be used. The SQL databases supported by default are MySQL/MariaDB and PostgreSQL, and some "big data" databases exposing a SQL interface such as PrestoDB and Apache Spark. The resourcer package can be extended to new DBI-compatible databases.

- NoSQL database resource which can be read using connectors from the nodbi package. Only the MongoDB database is supported for now.

- Command-based computation resource. The resourcer package can handle commands to be executed in the local shell and on a remote server through a secure shell (SSH) connection.

**Interacting with R Resources.**   As the ResourceClient is simply a connector to a resource, its utility is enhanced by a range of data conversion functions that are defined by default:

- R data.frame, which is the most common representation of tabular data in R. A data frame, as defined in R base, is an object stored in memory that may be not suitable for large to big datasets.

- dplyr tbl, which is another representation of tabular data provided by the dplyr package that nicely integrates with the R interface for databases: filtering, mutation and aggregation operations can be delegated to the underlying SQL database, reducing the R memory and computation footprint. Useful functions are also provided to perform joint operations on relational datasets.

In the case when the resource is a R object, the R data file ResourceClient offers the ability to get the internal raw data object. Then complex data structures, optimized for a specific research domain can be accessed with the most appropriate tools.

When the resource is a computation service provider, the interaction with the resource client will consist of issuing commands/requests with parameters and getting the result from it either as a response object or as a file to be downloaded.

The purpose of the resourcer package is definitely not to substitute itself for the underlying library; rather it is a general approach that facilitates the access to the data and service resources in the most specific way.

**Extending R Resources.**   Thanks to its modular and dynamic architecture, the *resourcer* package can easily be extended to:

- new file locations (see for instance the *s3.resourcer* R package which connects Amazon Web Service S3 file stores),

- new file readers (see Section 8 in our online book to see how VCF files are read in the *dsOmics* R package),

- new DBI-compatible databases. For instance, using *bigrquery* R package, it would be easy to implement access to a resource stored in a Google's BigQuery database.

- new domain specific applications which would expose data extraction and/or analysis services. The only requirement is that an R connection API exists for the considered resource application.

**Resources with DataSHIELD/Opal.**   DataSHIELD infrastructure is a software solution that allows simultaneous co-analysis of multiple data sets stored on different servers without the need to physically pool data or disclose sensitive information. DataSHIELD uses Opal servers to perform such analyses.

At a high level DataSHIELD is set up as a client-server model, each server housing the data for its corresponding study. A request is made from the client to run specific functions on the data held in the remote servers and that is where the required analyses are actually performed. Non-sensitive and pre-approved summary statistics are returned from each study server to the client where they can then be combined for an overall analysis. An overview of what a multi-site DataSHIELD architecture would look like is illustrated in **Fig 1**.

The current limitation of this infrastructure is that the data assignment operation consists of extracting the dataset from the data repository (the primary storage can be either a SQL

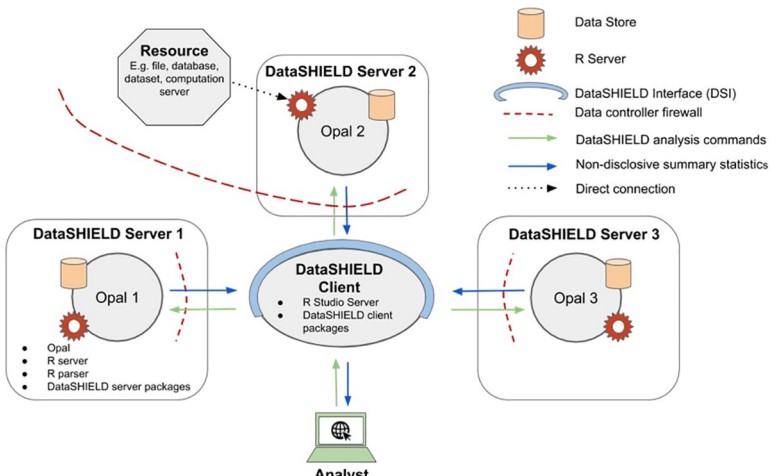

**Fig 1. A schematic diagram of a multi-site DataSHIELD infrastructure.** It includes one central analysis node (the client) and three data nodes (the servers).

database or a MongoDB database) and pushing it in the R server's memory as a data.frame object. This data assignment process consumes time, physical memory and applies only to datasets in tabular format. Although acceptable for small datasets (usually less than 10M data points), this infrastructure cannot be used for big data systems, complex data structures and existing computation facilities. In order to overcome these limitations, data access can instead be delegated to the R servers using the concept of resources. The DataSHIELD middleware (Opal) will be responsible for performing the assignment of a resource description as a ResourceClient object in the R server (as soon as the relevant permissions have been granted). The connection to data or computation resources is then readily usable by the DataSHIELD user. In terms of data access security, the resource credentials are not visible via the DataSHIELD client node, just like the individual-level data.

After converting a resource to a data.frame object, existing DataSHIELD analysis R packages can be used in a backward compatible way. New DataSHIELD R packages, such as dsOmics and dsGeo, described in this paper, can make use of the power of resources to apply DataSHIELD's data privacy preserving paradigm to large datasets, complex data structures and external computation facilities.

The integration of the resources concept in Opal consists of (1) dynamically discovering the different types of resources that can be handled by the associated R server, (2) providing an appropriate user-friendly graphical interface (GUI) for managing the resource descriptions and access permissions, and (3) assigning resource objects in the R server on user demand. The latest version of Opal (v 3.0) implements these capabilities, making resources accessible to the DataSHIELD framework.

## Results

**Table 1** describes the main features of our proposed framework that are mainly driven by DataSHIELD's capabilities. The table also shows the main advantages of disadvantages that can be found when using the *resources* in federated data analyses. Different features including scalability, disclosure prevention, deployment and future applications to other biomedical areas than genomics are discussed.

**Table 1. Main characteristics of the proposed infrastructure for privacy-protected federated data analyses with big data.**

| Feature | Capabilities / Advantages | Limitations / Disadvantages |
|---|---|---|
| Resources | Any data source or computation resource that can be accessed from R is made available in DataSHIELD. These include databases of any kind (SQL, NoSQL, distributed Big Data systems), most of the file formats, domain specific applications accessible through web services or remote commands etc. Applies to any scientific domain. | Resource URL design can be complex when resource options need to be specified. R is the required entry point, which complexifies the use of analysis algorithms in Python for instance. |
| Scalability | Scales with number of studies. The *resources* can interact with Apache Spark or Hadoop | Some advanced statistical techniques may not scale well with number of records. The processing power can sometimes suffer from latency. There is a need for investigation share computation over multiple processors. |
| R programming language | R is an open-source and heterogenous programming language. Interpreters for available for many operating systems. A wide community support its users. | Skills for other functional and statistical programming languages can be transferred to learn R. However not all of the analysts are familiar with other programming languages, which can raise some barriers of adoption. |
| Disclosure prevention | Several complementary features that provide privacy preservation are implemented in the DataSHIELD architecture. Just one of these is that data custodians/owners (not *analysts*) have complete control over a series of optional filters that dictate the potential disclosivity of the analytic output. As one example, this includes the minimum count acceptable in a non-empty cell of a contingency table. When rare observations are critical to analysis, data custodians can actively choose to relax the filters to enable meaningful analysis to be undertaken. One of the implications of the combined effect of several of the privacy preservation features is that analysts can never see, copy or abstract the individual-level data on the primary data servers. | Data protection is shared between existing computer systems and DataSHIELD. An existing computer system that has some poorly implemented secured network access and authentication may threaten the quality of disclosure prevention. In other words, as well as the active privacy preservation built into DataSHIELD software, it is also crucial that the hardware on which everything runs must satisfy good practice for conventional privacy protection. |
| Graphical User Interface (GUI) | The use of R studio and the specialised data warehouse software have a GUI. Analysis can be integrated with GUI packages, such as R Shiny. | An appropriate use of R Scripts, R notebooks and vignettes is helpful. DataSHIELD users then need to learn how to use these approaches before using DataSHIELD. The use of R Studio can make the learning curve less steep. |
| Applications to other biomedical areas than genomics | Any other specific data infrastructures available in public repositories such as images (OpenNeuro), transcriptomic or epigenomic data (GEO) can be accessed and analyzed in a distributed and privacy preserving way. Applications outside biomedical, health and social science are also entirely possible. | There are no obvious scientific or academic domains to which DataSHIELD could not in principle be applied. Particularly, if the aim is to facilitate the quantitative analysis of individual level data which are sensitive either because of ethico-legal or information-governance restrictions, or because of their intrinsic intellectual or commercial value. |
| Cost | As everything is based on open-source freeware, the costs are minimized for any organisations who wishes to adopt DataSHIELD. | While DataSHIELD development is substantively funded through grants, there is nevertheless a need to seek support from the user community (particularly large-scale users) to contribute to the core DataSHIELD provision: user training and support; on-line support materials; bug snagging and error correction; continuous testing of evolving code; preparing and smoothly undertaking new releases. |
| Time | From the user perspective, DataSHIELD is a time-effective tool as does not require iterative in-person communication between the analyst and the data holders. This is most evident in comparing the time commitment required for a standard consortium-based meta-analysis and that required by a centrally controlled study-level meta-analysis via DataSHIELD. The former requires the analysis centre to ask each study to undertake a series of specified analysis and return results (typically several rounds of analysis and return). In contrast the latter is controlled in real time as if one was working directly with the raw data. This can speed things up by orders of magnitude. | Like any specialised open-source software, DataSHIELD analysts, developers and installer need some time to adapt to the concepts of federated systems and disclosure limitations. |
| Server sharing | DataSHIELD supports multi-tenancy, to permit multiple users to share a DataSHIELD server. It also permits multiple servers to be deployed if needed. | DataSHIELD requires some specialised data warehouse software such as Opal. At the moment there is no white-paper that formally defines standards for further development in this area. |

(*Continued*)

**Table 1.** (Continued)

| Feature | Capabilities / Advantages | Limitations / Disadvantages |
|---|---|---|
| Documentation | DataSHIELD has a wiki that provides beginners training to any new DataSHIELD analysts and developers. Some more advanced tutorials are available on multimedia contents through YouTube. Documentation for the Opal specialised data warehouse software is available online.<br>The online book of this paper will be continuously being updated with more extensions (e.g. transcriptomics, epigenomics, imaging data, longitudinal data analyses, . . .) | DataSHIELD documentation needs to be more supportive to the DataSHIELD developers. An online forum is available for community engagement and support.<br>No advanced statistical techniques are explicitly taught, as it is assumed that any analyst planning to use DataSHIELD would already understand the theory and practice underpinning the analysis that is to be undertaken. However, in practice the DataSHIELD team knows that this is not always true and so some analytic theory is increasingly being presented when it is useful. |
| Deployment | Software solution packages are available for different hosting systems (includes a container-based option) and the resource concept can adapt itself to the existing hosting infrastructure.<br>It is advised that DataSHIELD should only be deployed in a setting in which all hardware and middleware systems satisfy conventional best practice for data management and privacy protection. Similarly, it is assumed that DataSHIELD will not be used if information governance or other restrictions already proscribe the particular analysis proposed. | It is important that there is at least a minimum baseline level of system administration knowledge on the part of the data owner and proper dimensioning of the hardware (especially when targeting multi-user, computation intensive usage). Nobody should be making sensitive data available for analysis via any mechanism–including DataSHIELD–if they do not have a proper understanding of their data systems or of the governance framework under which analysis is to be enacted. |

## Available resources extensions

We have extended the resources available at the *resourcer* package into different settings. These extensions as well as the current resources that can be accessed through the Opal servers are described in **Table 2**. So far, we can get data from different locations (Amazon Web Services, HL7 FHIR or Dremio), read other types of files which are specific in genomic studies (BAM, VCF and PLINK) and access data from other infrastructures such as GA4GH, a federated ecosystem for sharing genomic, clinical data [18] and EGA which is a permanent archive that promotes distribution and sharing of genetic and phenotype data consented for specific approved uses [19].

## Real data analyses

We illustrate how to perform privacy-protecting big data analyses using our proposed infrastructure. We have set up an Opal demo site (see Chapter 4 in our bookdown) to illustrate how to perform some basic analyses using DataSHIELD as well as how to deal with different *resources* for genomic and geographical data. These data are publicly available and can be accessed through DataSHIELD or using the URL available in the Opal site. The genomic example describes how to perform genome-wide association (GWAS). The geographic examples describe how to analyse information about journeys undertaken by specified individuals', the environment through which they travel and whether the journeys have an impact on the individuals' health. Chapter IV in our online book provides users with workflows and case studies for downstream analyses and visualizations.

**Genomic data analysis.** Bioconductor provides core data structures and methods that enable genome-scale analysis of high-throughput data in the context of the rich statistical programming environment offered by the R project [15]. We have created two packages to perform privacy-protecting federated genomic data analysis with DataSHIELD and Bioconductor. The *dsOmics* package contains the functions that are used on the server side where the actual analysis is implemented and which specify the privacy-protecting summary statistics that will be send back to the client, while the *dsOmicsClient* has the functions that are used on the client side, to control the commands that are send to the server side and to combine the received outcomes for pooled analysis applications.

**Table 2. Available resources at the *resourcer* R package and extensions for genomic data.**

| Type | Resource | Reference | R package | Use |
|---|---|---|---|---|
| File reader | R data | https://cran.r-project.org/ | resourcer | Any |
| File reader | Tidy data (.csv,.tsv, txt, . . .) | https://www.tidyverse.org/ | resourcer | Any |
| File location | S3 compatible file store | https://min.io/ https://aws.amazon.com/ | s3.resourcer | Any |
| Database | SQL | https://www.mysql.com/ https://mariadb.org/ https://www.postgresql.org/ https://prestodb.io/ | resourcer | Any |
| Database and Big Data analytics | SQL | https://spark.apache.org/ | resourcer | Any |
| Database | NoSQL | https://www.mongodb.com/ | resourcer | Any |
| Computation service | SSH | https://en.wikipedia.org/wiki/Secure_Shell | resourcer | Any |
| Domain specific | HL7 FHIR | http://hl7.org/fhir/ | fhir.resourcer | Patient's data |
| Database | SQL | https://www.dremio.com/ | odbc.resourcer | Any |
| File reader | VCF | https://en.wikipedia.org/wiki/Variant_Call_Format | dsOmics | Genomic |
| File reader | GDS | https://bioconductor.org/packages/release/bioc/html/gdsfmt.html | dsOmics | Genomic |
| File reader | BAM | http://samtools.github.io/hts-specs/SAMv1.pdf | dsOmics | Genomic |
| File reader | Bioconductor infrastructures (ExpressionSet, RangedSummarizedExperiment, MultiAssayExperiment, . . .) | http://bioconductor.org/ | dsOmics | Genomic |
| Computation service | PLINK | http://zzz.bwh.harvard.edu/plink/ | dsOmics | Genomic |
| Domain specific | GA4GH | https://www.ga4gh.org/ | dsOmics | Genomic and clinical |
| Domain specific | EGA | https://ega-archive.org/ | dsOmics | Genomic and clinical |

Genomic data can be stored in different formats. Variant Call Format (VCF) and PLINK files [20]. are commonly used in genetic epidemiology studies. In order to deal with this type of data, we have extended the resources available at the *resourcer* package to VCF files. We use the Genomic Data Storage (GDS) format which is designed for large-scale data management of genome-wide variants and can efficiently manage VCF files into the R environment. This extension requires the creation of specific *ResourceClient* and *ResourceResolver* classes. This extension is available in the *dsOmics* package (See Section 8 in the supplementary book). Briefly, the client class uses *snpgdsVCF2GDS* function implemented in *SNPrelate* to coerce the VCF file to a GDS object [21]. Then, the GDS object is loaded into R as an object of class *GdsGenotypeReader* from *GWASTools* package [22]. that facilitates downstream analyses such quality control of SNPs and individual data, population stratification and association analyses using *GENESIS* Bioconductor package [23].

**Fig 2** describes how GWAS may be performed using DataSHIELD client-side functions (i.e dsOmicsClient package). Basically, data (genomic and phenotypes/covariates) can be stored in different sites (EGA, GA4GH, https, ssh, AWS S3, local,. . .) and are managed with Opal through the resourcer package and their extensions implemented in dsOmics. The association analyses involving GWAS are based on fitting different generalized linear models (GLMs) for each SNP that can be performed using a base DataSHIELD function (ds.glm). The difference between standard data analysis and that done by DataSHIELD is that the analysis is performed

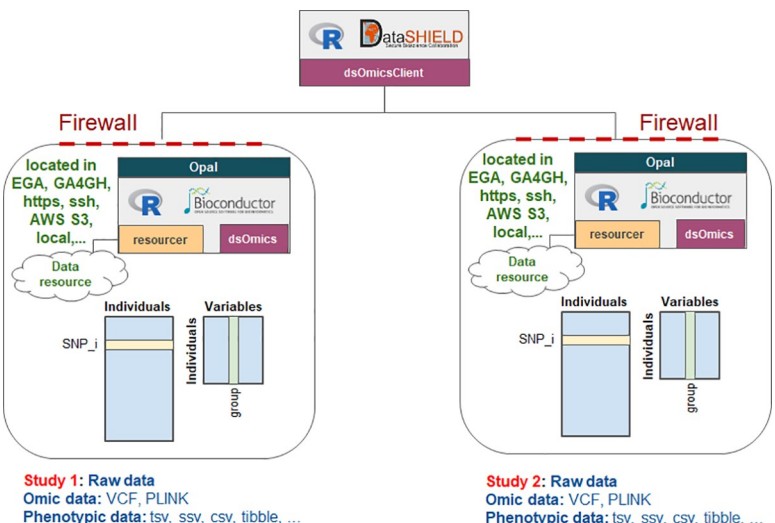

**Fig 2. Scheme of DataSHIELD implementention of genomic data analysis.** The *dsOmics* package contains functions to perform non-disclosive data analyses of resources encoding genomic data that are managed within Opal using the *resourcer* package. Genomic data normally have two pieces of information, one corresponding to variants (e.g. SNPs) and another for phenotypic data (grouping variable, outcome, covariates, . . .). Both can be stored in different resources. BAM/VCF/PLINK for SNPs and text/csv file for phenotypes and covariates. This package should be installed in the Opal server along with their dependences. The package *dsOmicsClient* must be available in the client side and contains functions that allow the interaction between the analysis computer and the servers.

at the location of the data (e.g. the data nodes). No data is transferred from that location, only non-disclosive summary statistics. The set of analytical operations which can be requested to be performed at the location of the data, has been careful constructed to prevent any attempt for direct or inferential disclosure of any individual-level information. The R parser also blocks any form of arguments that are not allowed in DataSHIELD. For analytical operations which could potential yield results which are disclosive, for example if a small amount of data is being analysed, the operation will check if the results match the data protection policies of the location's data governance rules, before returning any results back to the client (e.g. the analysis node). If any of the protection rules are violated, the client does not receive any results but gets study-side messages with information about potential disclosure issues [2].

It should be noticed that repeatedly calling ds.glm function can be very time consuming when analysing thousands of SNPs which requires multiple iterations of calls over the network between the client and the server. In order to overcome this problem, we also implemented a federated meta-analysis approach that basically runs an independent GWAS at each server and then meta-analyse the results. GDS data at each server are analyzed using GWASTools and GENESIS Bioconductor packages that allows to perform GWAS very quickly. Once the study-specific estimates and standard errors generated by the analyses undertaken on each server have been returned to the client, they can be combined using whatever meta-analysis approaches—and whatever R meta-analysis packages—the user may choose. This methodology has some limitations when data are not properly harmonized (e.g. genotyping in different platforms, different VCF versions,. . .). In order to overcome this problem, data format validation can also be performed by the analyst using DataSHIELD functions. In genomics, this can be achieved by first doing imputation and then solving issues concerning genomic strand and file format [24].

GWAS can also be performed using programs that are executed using shell commands. This is the case for PLINK, one of the state-of-the-art programs to run GWAS. Resources also

allow the use of secure SSH service to run programs on a remote server accessible through SSH containing data and analysis tools where R is just used for launching the analyses and aggregating results. This feature allows us to create functions to analyze data using specific shell programs. Section 9 in our online book describes how the PLINK program can be used to perform GWAS. In this case, the resource describes that access is given via SSH, the credentials required to connect, and the commands that can be run (of which one is plink).

We would like to emphasize that with DataSHIELD, analysis is performed at the location of the data. The data provider has full control over what information is transferred from their location to the location of the analyst by setting filters for a number of disclosure traps. This means that the results returned to the analyst can be carefully created to be non-disclosive, and match the policies of the data provider's data governance rules.

**Geographic Information System (GIS) and spatial analysis.**   The R packages *rgdal*, *rgeos* and *sp* provide core data structures and methods that enable analysis of geospatial data in the context of the rich statistical programming environment offered by the R project. We have created two packages to perform privacy-protecting federated GIS data analysis with DataSHIELD and these packages. The *dsGeo* package contains the functions used on the server side to assure privacy-protecting analyses, while *dsGeoClient* has functions that command the data analyses from the client side and enable integration across studies.

The *resourcer* package allows large geospatial datasets to be handled. These include data derived from standard storage systems, such as relational databases, making use of existing sp data structures such as *SpatialPoints* and *SpatialLinesDataFrame* among others. These types of data are the core of geospatial analysis in R, allowing users to work with geometries and their descriptive attributes. For example, we might want to know that a GPS trace corresponds to someone who is 45 years old, or that a region defined by a polygon has a particular air pollution level. As described in the methods section, resources can be extended to any type of data that can be managed within R. Here we describe how to extend the resources to the case of analysing Geographic Positioning System (GPS) traces and other geolocation data, combined with phenotypic data.

Section 13 in our online book illustrates how to perform a realistic analysis of GPS traces and geolocation data. In this example, building on the work of Burgoine et al., [25] we consider daily commutes captured as GPS traces by 810 individuals in the eastern suburbs of London. We also have data on the location of 6100 fast food or takeaway outlets in the same area. Further data are available, relating to each individual's Body Mass Index (BMI), age, sex, total household income, highest educational qualification and smoking status. These data therefore allow us to test the association between exposure to takeaway food on a commute and on individual's BMI. We illustrate how the tools available in the dsGeo package allow this question to be addressed.

We created three resources in Opal that contain the GPS journey data (*SpatialLinesDataFrame*), the locations of the food outlets (*SpatialPointsDataFrame*) and the phenotypic data (*data.frame*). These data can be manipulated to give a measure of exposure to the food outlets for each individual. Our package allows us to transform the point data denoting food outlets into buffer regions surrounding each point. The idea is that if an individual's GPS trace falls within this buffer then we can say that the individual is 'exposed' to that food outlet. Thus, we find the intersection of each individual's trace with each buffered region to obtain a vector of the intersecting buffers for each individual. This is further processed into a count of 'exposures' per individual. Finally we can run GLMs using the DataSHIELD base function *ds.glm()* to test the association between BMI and food outlet exposure, correcting for potential confounding factors such as income.

## Availability

We release resourcer R package under the open-source LGPL ($\geq$ 2.1) license. The source code can be downloaded from CRAN and our public repository https://github.com/obiba/resourcer. The server site (e.g. Opal) packages dsOmics and dsGeo are R packages under the open-source MIT and GPL3 licenses, respectively. They can be installed from GitHub https://github.com/isglobal-brge/dsOmics and https://github.com/tombisho/dsGeo. The client site R packages dsOmicsClient and dsGeoClient are also publicy available at GitHub repositories https://github.com/isglobal-brge/dsOmicsClient and https://github.com/tombisho/dsGeoClient. The "resources" extensions of GA4GH and EGA are available in the dsOmics package. R code describing how to perform the genomic and geographical data analyses using our proposed infrastructure are available in our bookdown https://isglobal-brge.github.io/resource_bookdown. Data used in this bookdown are publicly available and can be accessed through DataSHIELD or using the URL available in the Opal site https://opal-demo.obiba.org (user: administrator, password: password, see https://isglobal-brge.github.io/resource_bookdown/opal.html#opal-demo-site). Users may report bugs or other issues via the "Issues" tab at GitHub repositories or using the online forum at https://datashield.discourse.group/.

## Future perspectives

### DataSHIELD

The development of the new "resources" component of the OBiBa middle-ware that underpins the DataSHIELD platform has profound implications for the future of the overall DataSHIELD project. It has greatly relaxed constraints on the source, format and volume of the data that can be ingested into a DataSHIELD session while continuing to provide full flexibility in terms of the capacity to enact active disclosure control at a level appropriately tailored to the context of the particular analytic problem at hand. This has already allowed us to embark on exploring extensions of functionality to encompass some of the most widely used classes of contemporary research data, for example, Omic data and images. In tandem with recent advances we have made in learning how to simplify the extension of functionality in any field provided that a relevant R-package already exists, DataSHIELD is now on the cusp of being able to provide a generic easily usable and simply extendable approach to truly federated analysis across many science and technology domains. This will have a myriad of applications in academic research, commercial settings and health & social care systems. The ability to finely tune disclosure controls in a manner that can only be modified by the data custodian will make this approach particularly attractive for anybody wishing to work with data that are sensitive. This not only includes human (or other) data subject to the appropriately stringent requirements of contemporary law, ethics and broader frameworks for data governance, but also encompasses data that are sensitive for reasons of intellectual property investment or commercial value. However, if disclosure controls are set to be very permissive or off, the convenience, flexibility and potential extendibility of the system could ultimately make this an approach of choice for any federated analysis even if the data are not especially sensitive.

In light of the development of resources, it is envisaged that in the future there will be at least three flavours of DataSHIELD that will vary in their convenience of use, flexibility and ease of extension of functionality: (1) **All disclosure checks set to off**. This will permit maximum flexibility for analysing non-sensitive federated data and easily extendable by adding new functionality; (2) **Disclosure checks on but minimal**–*e.g.* preventing users from seeing or copying the individual level data but avoiding restrictions on the analyses themselves. This flavour is likely to be most useful when data from a large research platform cannot physically be

shared but the analysis required is based on data objects that are fundamentally privacy-protecting such as Omics data or images based on internal scanning; (3) **Full disclosure checks**; this will be equivalent to the default situation that applies now.

Now that it is possible for DataSHIELD to work much more readily with large/big datasets, we anticipate that DataSHIELD and the *resourcer* R package will offer functions based on the tidyverse as well as base-R. For example, this will include the option of using *dplyr* R package for operating on tabular datasets (see **Interacting with R Resources** section) and on tibbles as well as standard R data-frame objects. A resources integration improvement might therefore be to use the *dplyr* API for delegating as much as possible data filtering and mutating to the underlying data storage system (e.g. databases exposing a SQL query interface). This extension is already being explored.

## Parallel computing

Working effectively with large data may also require programming practices that match the available computer hardware infrastructure, both processing and storage. R is efficient when operating on vectors or arrays, so a pattern used by high-performing and scalable algorithms is to split the data into manageable chunks and to iterate over them. Chunks can be evaluated in parallel to gain speed. There are several R packages that can be used to this end (parallel, foreach, . . .) as well as *BiocParallel* Bioconductor package that facilitates parallel evaluation across different computing environments while allowing users from having to configure the technicalities. DataSHIELD analysis is parallelized by design (i.e. each server is working independently of the others). Therefore, for a server instance, the best approach is to use data structures and analysis tools that perform computations efficiently. This is the strongest point of the resources as we have done with the integration of dplyr and Bioconductor packages as well as those that can be implemented using, for instance, sparklyr [26].

## Omic and geographical data

We have provided some of the functionalities offered by Bioconductor and R packages in the DataSHIELD context that allow to analyse genomic and geographical data. These packages are extensive and more work is needed to repackage a more complete range of operations available in a privacy-protecting way. For instance, ds.Omics can be easily be extended to other omic data analyses such as differential gene expression analysis of methylation data analyses using the same strategy as the one used for genomic data. Visualisation can add great value and will be covered when being implemented in a privacy-protecting way. The *dsBaseClient* and *dsBase* packages already contain functions for privacy-protecting plots such as heatmaps and scatter plots. The privacy of data is protected in these cases by effectively blurring the data or by removing outlying points. These techniques could be adapted, for instance, to allow geospatial data to be visualised in a privacy-protecting way.

## Data cataloging

The next step of the resources integration in Opal is to make their meta-data findable to a researcher: exposing the data dictionaries, annotated with taxonomy or ontological terms, would benefit the research community when looking for datasets for a research question. OBiBa software application suite provides both Opal, the data repository (or data integration system, using the resources) and Mica [9], the data web portal application. Mica operates by extracting from Opal the dataset dictionaries to build a searchable data catalogue, with basic summary statistics and by allowing the submission of data access requests. Resources registered in Opal should be made visible from Mica as well.

## Other applications

We would like to highlight that there are dozens of disciplines other than genomics and geospatial that could also benefit from our infrastructure. For instance, extending the resources to other settings such as neuroimaging by using libraries from Neuroconductor, a similar project to Bioconductor for computational imaging, would be an important advance in that field since data confidentiality may also be an issue. Also, it is worth noting that one of the main advantages of using the resources is that we do not need to move data from their original repositories which can present a serious problem when dealing with neuroimaging data [27]. Another area that can readily benefit from our new framework is artificial intelligence. Big data and machine learning have applied innovatively many advanced statistical methodologies such as deep learning which is driving the creation of new and innovative clinical diagnostic applications among others [28]. The current trend is to include machine learning algorithms within Cloud capacities in different biomedical problems [29–31]. Our framework can interface with "Apache Spark", a fast and general engine for big data processing [32], through the sparklyr R package that will allow the use of different machine learning algorithms for big data.

## Acknowledgments

We would like to thank the software developers and scientific staff who have worked on Opal. We also acknowledge support provided by Maelstrom Research partners and invaluable feedback we have received from Opal users over the years.

## Author Contributions

**Conceptualization:** Yannick Marcon, Juan R. González.

**Formal analysis:** Tom Bishop, Xavier Escriba-Montagut, Juan R. González.

**Funding acquisition:** Paul Burton, Juan R. González.

**Resources:** Yannick Marcon.

**Software:** Yannick Marcon, Tom Bishop, Xavier Escriba-Montagut, Juan R. González.

**Supervision:** Juan R. González.

**Validation:** Demetris Avraam, Patricia Ryser-Welch, Stuart Wheater.

**Visualization:** Demetris Avraam, Xavier Escriba-Montagut, Juan R. González.

**Writing – original draft:** Yannick Marcon, Tom Bishop, Paul Burton, Juan R. González.

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
