## [Decision Letter · Decision Letter 0]

2 Oct 2020

Dear Dr Gonzalez,

Thank you very much for submitting your manuscript "Orchestrating non-disclosive big data analyses of data from different resources with R and DataSHIELD" for consideration at PLOS Computational Biology.

As with all papers reviewed by the journal, your manuscript was reviewed by members of the editorial board and by several independent reviewers. In light of the reviews (below this email), we would like to invite the resubmission of a significantly-revised version that takes into account the reviewers' comments.

Specifically, please simplify and make the main manuscript clearer based on reviewer's suggestions.

I didn't find a link to the code repository. Please provide it in the revised version.

We cannot make any decision about publication until we have seen the revised manuscript and your response to the reviewers' comments. Your revised manuscript is also likely to be sent to reviewers for further evaluation.

Sincerely,

Dina Schneidman

Software Editor

PLOS Computational Biology

Reviewer's Responses to Questions

**Comments to the Authors:**

Reviewer #1: Reproducibility report has been uploaded as an attachment.

Reviewer #2: Abstract: This abstract is too wordy and uses long sentences.

Example:

Therefore, big data analyses should ensure appropriate levels of security and privacy, be rigorous with the application of the data confidentiality regulations and address the choice between central data warehousing and the distributed (federated) analysis of data that remain ‘in-house’.

Why do we need "therefore"? What is "appropriate"? Why should analyses "address the choice"?

I can understand the basic objective in two sentences: Integrative analysis of multiple large datasets is a common objective in research [here you might focus on computational biology and use that term instead]. When data components of an integrative analysis are managed in distinct and independent systems, steps must be taken to ensure that confidentiality requirements are satisfied at all stages of the analysis.

With one more sentence you could describe key principles and tools to be described in the paper, and the abstract is done. For example, you could mention that DataSHIELD and Obiba are a decade old and that your work enhances these. Note that this reader has never heard of either of these projects, and that there is a cybersecurity company in Arizona USA called Datashield. This could be confusing, and you may want to clarify that Obiba is an open source project in the domain of bioinformatics, and that DataSHIELD is a collection of R packages usable by users of Opal.

Text: The first paragraph repeats much of the abstract; this should be avoided. The phrase "the Opal data warehouse" is used without explanation. I have never heard of this.

After browsing around obiba and looking at the three figures, I have a sense of what this project is about. The "book" https://isglobal-brge.github.io/resource_bookdown/ is nicely done and is a good supplemental resource for the paper. The figures of the paper give a vague sense of the collections of components in scope, and show "firewalls". But there is no clear depiction of how "non-disclosure" is achieved or guaranteed.

The paper suffers from the lack of tables that would help readers to understand capabilities and limitations in comparison to other frameworks. The paper also fails to mention GA4GH (Global Alliance for Genomics and Health), that aims to develop standards in this domain. See ga4gh.org, where all the main toolkits would seem to intersect with items described in the paper.

In summary, the paper reads like a mix between advertising and user manual, and mentions but really does not illuminate connections to computational biology practices. The tools appear well-motivated and well-documented, and may well deserve broad adoption. But this paper as written does not provide a strong argument for engaging with an inevitably complex environment. The basic principles can be articulated simply, but the proof that they are implemented in the system described here, and a demonstration that this implementation should be used by anyone who shares its objectives, will take more work and a more self-critical stance.

Reviewer #3: We were delighted to read about the datashield resources feature and think it is a great innovation that needs to get large attention from the scientific community. Datashield changes analysis practice by bringing analysis to the data, instead of first centralizing data, and does so in a practical usable way for researchers via the R statistical language. Thus, it provides a solution to the increasing difficult challenge of enabling pooled analysis of data from multiple centers and even countries for sensitive data which has become more challenging because of uncertainties around GDPR etc. However, datashield was limited to only the methods implemented by the datashield team and R who, while doing a great job, cannot be expected to address all analysis needs. The resources approach enables statististicians/bioinformaticians to create bespoke analysis pipelines and distribute them to centers automatically and provide access controls without needing to have much local analyst work (in contrast to sending around a cookbook and then requiring a local statistician to execute by hand).

Below we have comments and suggestions:

===

Overall we have the following concerns:

1.

The manuscript is in places uses many more words then necessary. This proza sometimes makes it hard to follow. So we would urge the authors to shorten and simplify the text where possible. Also carefully check for typos, in particular on the nice online book.

2.

One of the core features should be that data providers can precisely access control. However, it is not clearly described but based on test we believe it is actually quite simple. We think you should make this explicit from the start, i.e., in abstract, and also deserves a short alinea to describe how it works in practice

3.

While the authors give nice examples on analysis (actually more than needed), one thing that we found missing is how the authors believe complex analysis protocols will be distributed from central analysis site to local data providres. Because in case of sensitive data, we expect data providers to want to limit access to only one resource, i.e. a particular analysis procedure. Then the question is: how will such procedures be distributed from a central analysis site (i.e. lead of a consortium analysis) to all connected sites (i.e. data providers) without needing large local expertise. We believe that for example for distributed meta analysis such process should be seamless (otherwise still local expertise is needed to operate)

4.

Finally, in our own experience the hosting environment of the participating data providers can be quite heterogeneous. You might want to discuss how you expect to deal with a resource based analysis in for example a network of 20 data providers and how you would expect the analysis to accommodate these differences (without having to address these all via the central analysis lead).

===

Some text was hard to follow, and you might want to rephrase

5.

“Such analysis not only demands scalable methods and

appropriate mathematical models but must also maintain consistency with fundamental principles of

sound data management and data sharing that are common across all areas of health, social and

bioscience. Therefore, big data analyses should ensure appropriate levels of security and privacy, be

rigorous with the application of the data confidentiality regulations and address the choice between

central data warehousing and the distributed (federated) analysis of data that remain ‘in-house’.

“

6.

“This

requires data generators to physically transfer data [propose you add: to a central analysis server] to make them accessible to analytic users.”

7.

I got lost in the Methods section. It would greatly help if you give a short introduction, naming the main elements and how they fit together.

8.

“We define “resource” this data storage or computation access description”. I think you want to be more explicit, because apparently ‘resource’ is a URL that can denote file handle, data access protocol, or execution of some script.

===

Furthermore, we have minor suggestions that you can ignore but may be of use to the authors:

You state on page 1 that data has to be physically transferred to the warehouse. There are several tools that federate analysis to overcome this issue so you might want to rephrase. For example: https://www.dremio.com/.

Resources credentials are fixed and managed by Opal which implies you do not have a refresh token or something like that, that will expire over time. The policy decision and enforcement is now located in the DataSHIELD engine. This means that the resource owner can not decide anymore if someone has access to the resource. Audit logging at the resource side is hard to do this way.

If the computational resource needs to do a lot, you need to program that either in the ResourceExtension or as given commands.

When you program an extension you are dependent on the person's choices regarding the interface he/she exposes and when you give it as parameters in the resource you need to parse it in the resourcer package..

For example, how do you prevent malicious SQL injection? We would expect that you would not open up such resources to the external source but only a pre-packaged analysis.

You are still bound to the limitations of R in terms of memory and CPU usage when you want to correlate data in R against the data that is available in the resource. You need to either push the data you want to correlate against into the resource or extract it from the resource in the R-environment.

What is going to happen when you want to finish the analysis on another moment. Or the analysis is taking days how do you retrieve the result.

When the analysis is taking a lot of resources in terms of memory and CPU how do you limit this per DataSHIELD-user? Who is responsible for this, DataSHIELD or the resource owner?

The resource owner needs to implement a way to be easily accessible for the resourcer-package. Especially when you want to run more complex jobs it usually takes a complex interface to work with.

In the shell and ssh resource the Opal administrator is managing the actions that may be performed by the resource handler (researcher in general). When you host Opal on different infrastructure than the resource and is managed by someone who is not in charge of managing the resource the list of possible commands can be freely added to the resource. This possible allows

On page 7 and 12 you state that the disclosure control on big data is often more relaxed. In practice we do not encounter this relaxation. We are very keen on analysing data outside our facility but are bound to the current contracts regarding data transfer or access agreements which take a lot of time to arrange.

How can we make sure when you offer more open data in a way that it will be legally feasible to access the data without signing a data transfer or large access agreement?

In other words is it practically feasible to do analysis in this way regarding the juridical implications?

Maybe less dependencies in the resourcer package?

When the interface of a dependency is changed the package needs to be changed as well.

Think of a way to delegate the authorisation and authentication back to the resource owner. We would propose to change the way of passing credentials in the resource.

Is it possible to restrict the usage of certain resources? For example, prohibit the use of resources.

It would be nice to have some way of enforcing the metadata which is tight toi the data to be correct. This is often the problem with longitudinal data that metadata over all columns should be correct. It is likely that this also yields for big data structures that are analysed in a pooled manner. How do you handle multiple versions of VCF for example?

Book 16.3 is not finished

**Have all data underlying the figures and results presented in the manuscript been provided?**

Reviewer #1: None

Reviewer #2: **No: **It seems possible that the data related to Figure 3, or described in "Differential Gene Expression Analysis" is all available, but the paper does not provide explicit links, and it is not clear whether any reader would need to register or be authorized to interact with such data.

Reviewer #3: Yes

PLOS authors have the option to publish the peer review history of their article (what does this mean?). If published, this will include your full peer review and any attached files.

Reviewer #1: **Yes: **Anand K. Rampadarath

Reviewer #2: No

Reviewer #3: No
---

## [Decision Letter · Decision Letter 1]

17 Mar 2021

Dear Dr Gonzalez,

We are pleased to inform you that your manuscript 'Orchestrating privacy-protected big data analyses of data from different resources with R and DataSHIELD' has been provisionally accepted for publication in PLOS Computational Biology.

Best regards,

Dina Schneidman

Software Editor

PLOS Computational Biology

Reviewer's Responses to Questions

**Comments to the Authors:**

Reviewer #1: The Reproducibility report has been submitted as an attachment.

Reviewer #2: none

Reviewer #3: Thank your for this work and precisely described responses and changes made.

**Have all data underlying the figures and results presented in the manuscript been provided?**

Reviewer #1: Yes

Reviewer #2: Yes

Reviewer #3: Yes

PLOS authors have the option to publish the peer review history of their article (what does this mean?). If published, this will include your full peer review and any attached files.

Reviewer #1: **Yes: **Anand K. Rampadarath

Reviewer #2: No

Reviewer #3: No

---

## [Editor Report · Acceptance letter]

25 Mar 2021

PCOMPBIOL-D-20-01312R1 

Orchestrating privacy-protected big data analyses of data from different resources with R and DataSHIELD

Dear Dr Gonzalez,

I am pleased to inform you that your manuscript has been formally accepted for publication in PLOS Computational Biology. Your manuscript is now with our production department and you will be notified of the publication date in due course.

With kind regards,

Katalin Szabo
